# Neurological Impact of Respiratory Viruses: Insights into Glial Cell Responses in the Central Nervous System

**DOI:** 10.3390/microorganisms12081713

**Published:** 2024-08-20

**Authors:** Valentina P. Mora, Alexis M. Kalergis, Karen Bohmwald

**Affiliations:** 1Instituto de Ciencias Biomédicas, Facultad de Ciencias de la Salud, Universidad Autónoma de Chile, Santiago 8910060, Chile; valenpaz1999@gmail.com; 2Millennium Institute on Immunology and Immunotherapy (MIII), Facultad de Ciencias Biológicas, Pontificia Universidad Católica de Chile, Santiago 8331150, Chile; akalergis@bio.puc.cl; 3Departamento de Endocrinología, Facultad de Medicina, Pontificia Universidad Católica de Chile, Santiago 8331150, Chile

**Keywords:** central nervous system, respiratory viral infections, glial cells, microglia, astrocytes

## Abstract

Respiratory viral infections pose a significant public health threat, particularly in children and older adults, with high mortality rates. Some of these pathogens are the human respiratory syncytial virus (hRSV), severe acute respiratory coronavirus-2 (SARS-CoV-2), influenza viruses (IV), human parvovirus B19 (B19V), and human bocavirus 1 (HBoV1). These viruses cause various respiratory symptoms, including cough, fever, bronchiolitis, and pneumonia. Notably, these viruses can also impact the central nervous system (CNS), leading to acute manifestations such as seizures, encephalopathies, encephalitis, neurological sequelae, and long-term complications. The precise mechanisms by which these viruses affect the CNS are not fully understood. Glial cells, specifically microglia and astrocytes within the CNS, play pivotal roles in maintaining brain homeostasis and regulating immune responses. Exploring how these cells interact with viral pathogens, such as hRSV, SARS-CoV-2, IVs, B19V, and HBoV1, offers crucial insights into the significant impact of respiratory viruses on the CNS. This review article examines hRSV, SARS-CoV-2, IV, B19V, and HBoV1 interactions with microglia and astrocytes, shedding light on potential neurological consequences.

## 1. Introduction

The central nervous system (CNS) comprises two main cell types: neurons and glial cells [1]. Within glial cells, we can find microglia, astrocytes, and oligodendrocytes [1,2]. Even though neurons are responsible for signal transmission and information processing, glial cells play important roles in the brain [1,2]. Among these roles, we can find nervous system development and activity, synapse formation, neuronal migration, myelin generation, and homeostasis [1,2]. Also, they have a secondary but still as important role as mediators of immune responses in the brain [3].

Respiratory viruses are a general concern in the population, causing high mortality rates [4]. These viruses can cause bronchiolitis and pneumonia, affecting mainly young children and older adults [4]. Some of these respiratory viruses are the human respiratory syncytial virus (hRSV), severe acute respiratory coronavirus-2 (SARS-CoV-2), influenza virus (IV), human parvovirus B19 (B19V), and human bocavirus 1 (HBoV1) [5,6,7]. The respiratory manifestations caused by these viruses generate an inflammatory state, which has been shown to induce a hypoxic state that can exacerbate the infection [8]. This hypoxic response is associated with hypoxia-inducible factor 1/2 alpha (HIF-1/2α) signaling [8].

Besides causing respiratory complications, these viruses are also associated with implications for the CNS, where they can alter signaling pathways [4,7,9,10].

These alterations within neurons and glial cells translate to clinical signs, including fever, seizures, convulsions, encephalitis, encephalopathy, delirium, and abnormal behavior [4,10]. These clinical manifestations have led to long-term complications, such as schizophrenia, autism spectrum disorder, and mood disorders [11]. The mechanism by which these viruses can cause these consequences on the CNS has yet to be fully elucidated. However, research associated with the infection of CNS cells, such as neurons and glial cells, by these viruses could give a better understanding of how respiratory viruses affect the CNS.

Here, we will review the implications of hRSV, SARS-CoV-2, IV, B19, and HBoV1 on the CNS, focusing on glial cells such as microglia and astrocytes, considering their role in brain homeostasis and immune responses in the CNS.

## 2. Glial Cells in the Central Nervous System

As stated above, glial cells comprise an important part of the CNS [1,2]. Even though their complete properties remain unknown, they carry out significant roles associated with CNS maintenance, surveillance, and defense [1,2,3,12]. Within glial cells, specifically in pathology, microglia and astrocytes carry out more immune-associated roles, whereas oligodendrocytes are more associated with demyelination processes [12]. For this reason, we will focus on microglia and astrocytes in the following sections. Microglia, a myeloid cell in the CNS, are implicated in activating and maintaining type I immunity due to their role in cytokine secretion after infection [13,14,15]. They also express a wide range of toll-like receptors (TLRs) and molecules associated with their surveillance role [13]. Besides their contribution to tissue repair and remyelination, it has been shown that after exposure of microglia to Th_1_ or Th_2_ T cell supernatant, they can switch to different polarization states [13]. On the other hand, the primary immune role of astrocytes is the regulation of exchanges through the blood-brain barrier (BBB) [13]. Like microglia, they can also secrete cytokines, express major histocompatibility complex (MHC)-II, and modulate the immune response [13]. Moreover, communication between microglia and astrocytes allows regulation between both cells in injury or repair conditions [13]. Further characterizing these cells and their responses against a disturbance in the CNS would enable us to understand CNS responses better. This section will discuss the significant roles of microglia and astrocytes in the immune response of the CNS.

### 2.1. Microglia

Microglia are the most common tissue-resident myeloid cells in the CNS, even though they only comprise 10–15% of all glial cells in the CNS [14,15]. Microglia originate from erythromyeloid progenitors in the yolk sac and colonize the CNS during early neurogenesis using metalloproteinases [2,14,16]. This arrival in the CNS before neurogenesis and neuronal wiring processes had begun indicates a crucial role of microglia in correct brain development [17]. Moreover, it has been shown that microglia control neural precursor cell (NPC) proliferation through phagocytosis and promote their differentiation [17]. Microglia are also participants in gliogenesis processes, where they induce the differentiation of astrocytes and aid in the maturation and survival of oligodendrocytes [17].

Given this, microglia are vital players in neuronal survival, development, and brain homeostasis by communicating directly with neurons, astrocytes, and blood vessels [2,14]. Microglia can also constantly monitor the brain and modify their responses according to stimuli [18]. During normal and homeostatic CNS conditions, the microglia cell population is maintained in a stationary state [14,16]. This is characterized by slow and local proliferation and low expression of MHC-I, MHC-II, and co-stimulatory molecules, CD40 and CD86 [14,15,16,18]. However, during neuroinflammation triggered by an injury, an infection, or a neurodegenerative disease, microglia are recruited and undergo rapid proliferation, transitioning to an activated phenotype, accompanied by morphological changes, increased phagocytic activity, the release of reactive oxygen species (ROS), and the initial orchestration of an innate immune response characterized by the secretion of cytokines such as interleukin (IL)-6, IL-1, and tumor necrosis factor-alpha (TNF-α) [14,15,16,18]. During this process, microglia also increase the expression of MHC class I and II and co-stimulatory molecules, making them a tremendous antigen-presenting cell (APC) [18]. This allows interaction with T cells from the periphery, increasing the immune response [18]. Microglia also express TLRs, which are upregulated in the presence of neurodegenerative diseases [19]. Microglial TLRs were also stimulated after infection with Theiler’s encephalomyelitis virus (TMEV). They upregulated the expression of MHC-II and co-stimulatory molecules, further promoting the APC state of the microglia [20].

Lastly, it is important to mention that activated microglia tend to polarize towards two phenotypes, M1 and M2 [21]. M1 microglia have a pro-inflammatory profile associated with producing pro-inflammatory cytokines and chemokines, leading to inflammation and neuronal death [21]. On the other hand, M2 microglia have an anti-inflammatory profile related to neuroprotection, tissue maintenance, and repair [21]. However, this dichotomy in microglial profiles has also been classified as simplifying microglial responses [22,23].

### 2.2. Astrocytes

Astrocytes are the most abundant cell type in the CNS, representing around 20–40% of total cells in the CNS, outnumbering neurons [24,25,26]. Astrocytes originate from “radial glia”, which are neuroepithelial progenitors that differentiate towards astrocytes after the competition of the neurogenesis process and gain characteristics associated with astrocytes [2,27]. They translocate to the grey or white matter and can directly generate proliferating astrocytes or use oligodendrocyte progenitors as an intermediate [2,27]. Astrocytes can proliferate locally in the cortex and spinal cord, carrying out roles in the proliferation and migration of oligodendrocyte progenitor cells [27,28]. Other roles of astrocytes include brain homeostasis, neural development, and function by releasing neurotrophic factors [26,29]. They also regulate pH, blood–barrier formation and maintenance, neurotransmission, and the metabolic support of neurotransmitters [26,29]. Astrocytes can be recognized by their morphology and glial fibrillary acidic protein (GFAP) expression since this cell type exclusively expresses this molecule. After an injury or infection, GFAP is upregulated, which indicates an astrocyte’s reactivity [30,31].

Astrocytes can be classified into two main groups: fibrous astrocytes, in the white matter, and protoplasmic astrocytes, in the grey matter [32]. Because of this heterogeneity, astrocytes react differently during CNS injury or infection [24,33]. During pathological stimuli, astrocytes become activated, undergo morphological changes, and secrete ROS and pro-inflammatory cytokines and chemokines, such as TNF-α, IL-6, and IL-1β [34,35]. Astrocytes also express TLRs, both TLR2 and TLR4, which have proven to recognize and respond to LPS stimuli independently from microglia, secreting the cytokines previously mentioned [35]. This supports the idea that astrocytes can mount immune responses, which are critical in neuroinflammation processes [35].

It is important to mention that different phenotypes of astrocytes, such as A1 and A2 phenotypes, can have similar functions to those described in microglia [24]. A1 astrocytes are characterized by neurotoxic activity and secretion of the previously mentioned cytokines, causing loss of synaptogenesis and neuronal death. They are believed to participate in the pathogenesis of neurological diseases [24,34]. On the other hand, A2 astrocytes have been associated with neuroprotective activity, promoting synaptic repair, growth, and neuronal survival [24,34]. However, astrocyte responses are more flexible and disease-specific, making this classification controversial [22].

## 3. Respiratory Viruses and the Glial Cells of the CNS

As stated previously, respiratory viruses are an important health burden worldwide, being responsible for thousands of hospitalizations and deaths, mainly among young children, the elderly, and immunocompromised populations [4,36]. These viruses can cause many symptoms within the respiratory tract while also affecting other organs and systems, such as the CNS [4,7,9,10]. Neurological manifestations associated with respiratory viruses in the CNS include clinical signs such as fever, seizures, convulsions, encephalitis, and encephalomyelitis, among many others [4]. Even more, reports have indicated long-term consequences associated with viral respiratory infections and the CNS, such as abnormal behavior, depression, delirium, language learning impairment, anxiety, and even post-traumatic stress disorder [4,37]. Considering the role and importance of microglia and astrocytes in mounting immune responses in the brain [3], further characterization of the role of respiratory viruses in the CNS and their interaction with these glial cells can shed light on possible mechanisms and pathways used by the viruses to cause neurological manifestations. Here, we will review the impact of common respiratory viruses on these cells and how they could be associated with the neurological manifestations described above.

### 3.1. Human Respiratory Syncytial Virus

The human RSV (human orthopneumovirus) belongs to the genus *Orthopneumovirus*, the *Pneumoviridae* family, and the *Monogenavirales* order [4,38]. Its genome is an enveloped, negative-sense, single-stranded RNA molecule [38,39]. hRSV causes acute lower respiratory tract infections (ALRTI), with clinical manifestations ranging from rhinorrhea, cough, and respiratory distress to bronchiolitis and pneumonia [39]. hRSV affects mainly children and the elderly population [39]. This generates high hospitalization and mortality rates worldwide, making hRSV infection a vital disease burden [38]. hRSV infection has also been associated with the activation of HIF-1α, which has been seen to support the production of hRSV virions and lead to severe lung damage [8,40]. Other findings have shown that HIF-1α activation in hRSV infection does not associate with oxygen variations, suggesting that it is not caused by hypoxia [41]. On the other hand, studies have also shown that HIF-1α downregulates nucleolin expression, which is one of the hRSV receptors, possibly limiting viral replication [42]. Besides its pulmonary effect, it has been suggested that hRSV infection could lead to neurological manifestations, such as seizures, encephalopathy, and encephalitis, causing long-term consequences [4,39,43,44,45]. In this line, hRSV can also infect cells in the CNS, such as neurons, microglia, and astrocytes [46]. These CNS complications could also be mediated by changes associated with the hypoxic state generated after infection and the activation of HIF-1α. In this line, a study detected hRSV in the CSF of five children with encephalopathies. However, those with hypoxic encephalopathies were hRSV-negative [47]. This suggests that hypoxia might not play a role in the CNS manifestations of hRSV. However, further studies need to be conducted, mainly due to the different effects of HIF-1α on hRSV infection.

Data about hRSV’s receptor used to infect microglia and astrocytes is deficient. However, several receptors for the virus have been identified in other cell types, such as CX3 chemokine receptor 1 (CX3CR1), nucleolin, epidermal growth factor receptor (EGFR), insulin-like growth factor-1 receptor (IGF1R), heparan sulfate proteoglycans (HSPGs), and intercellular adhesion molecule-1 (ICAM-1) [48]. These receptors interact differently with hRSV to promote infection-1 [48]. CXCR1 binds to the CX3C motif on the glycoprotein (G) of hRSV in the lung, initiating infection [48]. On the other hand, nucleolin acts as a receptor for the fusion (F) protein of the virus in association with the internalization of the virus, where IGF1R, which also binds to the F protein, mediates the translocation of nucleolin to the cell membrane [48]. Similarly, EGFR also interacts with the F protein, promoting mucus secretion, while ICAM-1 contributes to airway inflammation [48]. On the other hand, HSPGs can interact with the G and F proteins to facilitate infection and viral attachment in vitro [48]. These receptors are expressed in microglia and astrocytes [48,49,50,51,52,53,54,55,56]. Even though studies specifically regarding these glial cells and receptors have not been conducted for hRSV, it is possible to suggest that hRSV could use these receptors to promote viral attachment and replication in glial cells (Table 1). Moreover, considering that these receptors mediate different routes and diverse effects of the virus, it would be necessary for further studies to assess the role of each receptor in microglia and astrocytes.

In regards to microglia, studies have shown an increase in the ionized calcium-binding adapter molecule 1 (Iba-1), a marker for microglia after hRSV infection, and a decrease in the neuronal nuclear (NeuN) protein, a marker for neuronal populations [57]. This suggests an association between hRSV infection and correct brain function, possibly through a mechanism associated with microglial activation [57]. This could be related to the M1 pro-inflammatory profile promoted by hRSV infection, as was shown by increased levels of IL-1β and inducible nitric oxide synthase (iNOS) (Table 1) (Figure 1) [58]. Moreover, it has been described that HIFs can regulate M1/M2 polarization states, influencing this M1 microglial activation [8].

Studies conducted on a microglia cell line have also shown a correlation between neuronal death and microglial activation due to the virus [59]. hRSV infection promoted an increased voltage-gated proton channel (Hv1), which is involved in ROS production and is selectively expressed in microglia [59]. This leads to an increase in ROS production and possible neuronal death [59]. This study also showed that hRSV infection or exogenous cytokine treatment induces axon extension and enlargement of the cell body of the remaining survival neurons [59]. hRSV infection also promotes the gene expression of TLR3 and RIG-I, which have previously been shown to induce cellular apoptosis (Table 1) (Figure 1) [59]. This suggests that hRSV infection in microglia results in neuronal damage, which could be inhibited or treated by blocking TLR3 and/or RIG-I expression. This neuronal damage could be directly linked to the clinical manifestations of the virus, both during the infection, such as seizures and encephalitis, and even in the long term.

Studies in astrocytes have shown an increase at 60 days post-infection (dpi) in GFAP expression and protein levels in the brain, which suggests that hRSV infection promotes astrocyte activation [46]. Increased IL-4, IL-10, and CCL2 also accompanied this [46]. Besides promoting a pro-inflammatory state within astrocytes, hRSV has been shown to increase the permeability of the BBB, as detected by an increase in Evans Blue extravasation to the brain of mice at 3 dpi [46]. Due to this, peripheral macromolecules can enter the brain and promote the pro-inflammatory state induced by the virus [46]. Considering the role of astrocytes in BBB maintenance, this suggests that hRSV can alter the correct function of astrocytes, promoting a pro-inflammatory state within the brain. Additionally, in vitro data showed that hRSV infection increased over time with augmented GFAP and nitric oxide (NO) production. Moreover, hRSV-infected astrocytes produced IL-4, IL-10, and TNF-α during the first hours of infection and IL-6 in late times (Table 1) (Figure 1) [46]. This pro-inflammatory state seen in astrocytes could also lead to neuronal death and a loss of synaptogenesis, promoting polarization towards an A1 profile in astrocytes.

Overall, even though the specific receptor used by hRSV to infect microglia and astrocytes has not been elucidated, the virus causes activation of microglia and astrocytes towards a pro-inflammatory profile on the brain, accompanied by the secretion of pro-inflammatory cytokines and neuronal death, which could be directly linked to long-term consequences in the CNS [46,57,58,59].

### 3.2. Severe Acute Respiratory Syndrome Coronavirus 2

SARS-CoV-2 belongs to the genus *Betacoronavirus*, the *Coronaviridae* family, and the *Nidovirales* order [60,61]. SARS-CoV-2 is an enveloped virus with a non-segmented, positive-sense, single-stranded RNA molecule [61]. Mild cases of SARS-CoV-2 infections are associated with fever, fatigue, headache, and diarrhea. However, in more severe cases, clinical manifestations such as dyspnea and respiratory failure tend to occur, leading to multi-organ failure and death [61,62]. SARS-CoV-2 was first detected in China, leading to the COVID-19 pandemic, which caused a gigantic global health crisis and significantly impacted healthcare systems [62].

Besides causing respiratory manifestations, SARS-CoV-2 has also been associated with neurological and psychiatric complications, such as headache, ischemic stroke, seizures, and encephalopathies [63]. SARS-CoV-2 is also responsible for neuropsychiatric complications [64]. However, the mechanisms underlying the development of these manifestations are unknown [64]. Hypoxia also plays a role in SARS-CoV-2 infection; HIF-1α accumulates in the lung epithelia after infection, and due to its role as a transcription factor for pro-inflammatory cytokines, it may play a role in the lung damage seen after SARS-CoV-2 infection [8,65,66]. Moreover, a hypoxic state titled “silent hypoxia” can be seen in COVID-19 patients because, even though the patient is hypoxemic, there are no visible symptoms indicating this state [8]. A possible hypothesis for this condition is that the lack of dyspnea generated in silent hypoxia is associated with neuronal damage [8]. SARS-CoV-2 can inflict this neuronal damage by directly infecting neurons of the limbic system or through a cytokine storm [8]. This suggests that hypoxia could be directly linked to the neurological and psychiatric disorders seen in SARS-CoV-2 patients. In this line, a study on non-human primates showed neuroinflammation associated with brain hypoxia after infection with the virus [67,68]. Moreover, histopathological brain analyses of patients who died of SARS-CoV-2 present signs of hypoxia-related injuries in the cerebellum and cerebrum [68].

SARS-CoV-2 can potentially affect the brain through systemic inflammatory processes involving the secretion of various cytokines [22,69]. However, some research has explored the virus’s direct interaction with the angiotensin-converting enzyme 2 (ACE-2) receptor in the brain. This receptor is primarily expressed by endothelial cells but is also found in neurons and glial cells [22]. Interestingly, even though studies have shown that HIF-1α promotes SARS-CoV-2 replication, the activation of this transcription factor also reduces the expression of ACE2 [8,66]. This suggests that another SARS-CoV-2 receptor could mediate the hypoxic state associated with HIF-1α, which promotes the pro-inflammatory state. In this line, in addition to ACE-2, other receptors such as the erythropoietin-producing hepatocellular (Eph) receptor, CD147, neuropilin-1 (NRP-1), the receptor tyrosine kinase AXL, and HSPGs have been identified for SARS-CoV-2 [70,71]. The potential disruption of the BBB, a protective barrier against foreign molecules, facilitates the virus’s ability to infect the brain [22,69]. This disruption allows the entry of pathogenic agents, triggering responses in astrocytes and microglia, among other cells, through various mechanisms [22]. Single-cell analyses have shown that ACE2 is rarely distributed within microglial cells [72]. However, this does not eliminate the possibility of SARS-CoV-2 infection since it has been demonstrated that this receptor is critical for the neuroinvasion caused by the virus [72]. On the other hand, CD147 has been co-localized with Iba-1 in LPS-treated mouse brains [73]. This suggests the virus could use the receptor under the neuroinflammatory process generated by SARS-CoV-2 infection. Lastly, microglia expresses HSPGs and the AXL receptor, suggesting it can work as a SARS-CoV-2 receptor in these cells (Table 1) [56,74].

In microglia, research indicates that SARS-CoV-2 infection induces immune cell accumulation and microgliosis within the brain, a phenomenon observed in post-mortem examinations of individuals who succumbed to COVID-19 [63,75]. This microgliosis is accompanied by CD8^+^ T cell infiltration and an up-regulation of programmed death 1 (PD-1) [76].

Considering this, studies in the immortalized human embryonic brain-derived primary microglia cell line (HMC3) showed that SARS-CoV-2 can directly infect these cells, generating pro-inflammatory responses associated with an M1 phenotype [63]. These responses were characterized by increased secretion of IL-1β, IL-6, and TNF-α, alongside a high RNA expression level of nitric oxide synthase 2 (NOS2), a marker for M1 activation (Table 1) (Figure 2) [63]. This M1 phenotype has also been associated with HIF-1α activation, suggesting a direct effect of hypoxia on microglial polarization [77]. This switch to an M1 polarization state is also related to the activation of JAK-STAT signaling, which, in turn, will continue to promote the pro-inflammatory state [75].

Even more, intranasal infection on K18-hACE2 transgenic mice, which express the human ACE2 receptor, showed viral RNA in the brains of SARS-CoV-2 infected mice, alongside colocalization at 6 dpi of the spike (S) protein of this virus and Iba-1 [63]. Microglial depletion in K18-hACE2 transgenic mice correlates with reduced expression of pro-inflammatory cytokines and chemokines, such as CCL2 and CCL5 [78]. This shows that SARS-CoV-2 can infect microglia in vitro and in vivo, promoting a pro-inflammatory state within the brain [63].

Besides causing a pro-inflammatory effect, SARS-CoV-2 infection of HMC3 cells has caused an induction of the gene expression of ER stress responses at 3 dpi, which continued to be seen at 6 dpi, alongside gene expression increments of apoptotic signaling pathways [63]. This shows that SARS-CoV-2 infection can induce cell death and apoptosis as a cytopathic effect, as was confirmed by Annexin V staining (Table 1) (Figure 2) [63].

Studies in vivo of microglial activation after SARS-CoV-2 exposure also show cell proliferation, an increase in TNF-α and IL-6, and an upregulation of nucleotide-binding domain, leucine-rich-containing family, and pyrin domain-containing-3 (NLRP3) inflammasome components, as demonstrated in K18-hACE2 transgenic mice and human monocyte-derived microglia [79,80]. This inflammasome activation was dose-dependent on the S protein through the ACE2 receptor and potentiated in the presence of α-synuclein. This protein aggregates during Parkinson’s disease (PD) progression [80]. This could suggest a possible role for SARS-CoV-2 infection in PD. In this line, it has been shown that NLRP3 inflammasome in microglia contributes to autophagy and pro-inflammatory responses due to the virus [75]. It is possibly linked to its association with NF-κB and the induction of MyD88 and TRIF [75].

Furthermore, infecting brain organoids with developing microglia showed a reduction in post-synaptic density, accompanied by an upregulation of genes associated with IFN, phagocytosis, and synapsis elimination (Figure 2) [64]. Neurodegenerative diseases have been associated with synaptic dysfunction and early synapse loss, which could suggest that the higher risk for neurological complications in SARS-CoV-2 infection, such as schizophrenia, could be related to microglial synapsis elimination (Table 1) [64].

On the other hand, previous studies with neuro-invasive viruses, such as the West Nile virus, have shown that CNS damage can be attributed to IFN-γ signaling in microglia [64,81]. This could indicate a similar association in the neuropathological manifestations of SARS-CoV-2 [64]. Considering this, single-cell transcriptomics also showed that the signatures adopted by glial genes overlap with the ones seen in neurodegenerative diseases [64]. Studies conducted on wild-type C57BL/6 mice have shown that an injection with the E protein mediates depression-like behaviors in mice and dysosmia, mainly through microglial activation and TLR2 activation (Table 1) (Figure 2) [82]. This suggests that TLR2 activation could be a possible therapeutic target to prevent SARS-CoV-2 depression-like sequelae.

Astrocytes, on the other hand, similarly to what is seen with microglia, undergo astrogliosis processes after SARS-CoV-2 infection, as could be evidenced by an increase in the plasma concentration of the GFAP marker in COVID-19 patients (Table 1) (Figure 2) [22]. The virus also alters the secretion of various proteins in astrocytes, which suggests a compromise of the BBB. As described, a similar situation could occur after hRSV infection [46,64]. This further supports the potential entry route of SARS-CoV-2 through a disruption of the BBB. BBB disruption has also been associated with hypoxia; an in vitro model of BBB disruption with endothelial cells and astrocytes showed that hypoxia reduces the expression of ZO [83].

Further supporting this, studies conducted on rhesus monkeys showed that SARS-CoV-2 can infect astrocytes, as was revealed by a colocalization of the N protein with GFAP^+^ cells [84]. Besides this colocalization, an increase in GFAP^+^ astrocytes was also noticed in the piriform cortex, suggesting its proliferation or possible translocation to the site (Table 1) [84]. Results in these animals also showed that the infection led to morphological abnormalities in astrocytes, in association with their role in the function of the BBB (Table 1) (Figure 2) [84]. This could be associated with the entry of peripheral cells into the brain during infection [84,85]. It is believed that SARS-CoV-2 can directly infect pericyte cells, which form the BBB alongside astrocytes; this may cause astrocyte death, which is directly linked to the disruption of the barrier [76]. Other studies have suggested that the virus crosses the BBB through non-specific endocytosis or indirectly through peripheral inflammation [76].

Studies conducted on K18-hACE2 transgenic mice and Syrian hamsters showed that SARS-CoV-2 was seen in the vascular wall and perivascular space, possibly infecting endothelial cells and supporting the hypothesis that this virus crosses the BBB [86]. Astrocytes from K18-hACE2 mice were significantly activated in the cortex and hippocampus, accompanied by increased levels of IL-6, TNF-α, and CCL2 in the brains of these mice [86]. This could further suggest that astrocyte activation could disrupt BBB in mice (Table 1) (Figure 2) [86]. Furthermore, after administration of Evans blue dye, leakage was evident in the cortex of the animals, particularly in Syrian hamsters, which exhibited more substantial damage than k18-hACE2 mice and destruction of basement membranes, as corroborated by Masson’s stain (Figure 2) [86]. It was also found that matrix metalloproteinase 9 (MMP9) levels were increased, which can degrade collagen IV and cause BBB breakdown (Figure 2) [86]. This also suggests that SARS-CoV-2 possibly damages the BBB by disrupting basement membranes via MMP9 [86]. It is important to note that no significant decrease in tight junctions was seen, which suggests that the virus crosses the BBB in a transcellular pathway [86]. However, other studies have found that an injection with the envelope protein causes a decrease in zonula ocludens 1 (ZO-1) expression in the cortex, hippocampus, and olfactory bulb of C57BL/6 mice [82]. All of this further supports SARS-CoV-2 entry through the BBB.

Even though studies have described ACE2 distribution in astrocytes, other studies in brain organoids have shown that SARS-CoV-2 infects astrocytes through the NRP-1 receptor, which is highly expressed in astrocytes (Table 1) (Figure 2) [72,87,88]. Furthermore, a siRNA knockdown of NRP-1 inhibited the infection of astrocytes by SARS-CoV-2, as confirmed by a reduction in the expression of the nucleocapsid (N) mRNA [88]. The knockdown of the ACE2 receptor could not be accurately assessed because of its low baseline expression in astrocytes [88]. Because of this, treatment of astrocytes with a neutralizing antibody against ACE2 was evaluated [88]. However, it did not significantly impair SARS-CoV-2 infection [88]. This shows that the NRP-1 receptor could be a significant focus of study to analyze the manifestations of SARS-CoV-2 in the CNS in association with astrocyte infection. Even more, studies conducted on brain samples from patients who died of COVID-19 showed undetectable levels of ACE2 mRNA, and once again, neutralizing the NRP-1 receptor inhibits SARS-CoV-2 infection on stem-cell-derived astrocyte extracts, confirming what was seen in studies with brain organoids [89].

Other possible receptors for SARS-CoV-2 in astrocytes include the Eph receptor, expressed in astrocytes, and CD147, co-localized with GFAP^+^ astrocytes in stroke mice [70,90]. Lastly, HSPGs and AXL, which are expressed by astrocytes, have also been identified as receptors for the virus (Table 1) [55,71,91].

Genetic analyses of the brain organoids infected with SARS-CoV-2 revealed a downregulation in calcium/calmodulin-dependent protein kinase II delta (CAMK2D), receptor tyrosine kinase erbB2 (ERBB2), complement C1q-like protein (C1QL), and synaptophysin-like protein 1 (SYPL1) genes associated with synapse function [88]. Even more, analyses of these brain organoids and the primary cell culture of astrocytes showed that several genes related to the type I IFN pathway were upregulated and activated [88]. Primary cultures of astrocytes also showed increased transcription levels of CXCL10, CXCL6, CXCL1, CXCL2, and IL-17, suggesting a pro-inflammatory phenotype [88]. Neuron cell death was also evaluated, revealing that these cells were SARS-CoV-2 negative. This meant that their death was not caused by direct infection but by changes in neighboring cells or the environment (Table 1) (Figure 2) [88]. This suggests that the pro-inflammatory environment generated by SARS-CoV-2 infection triggers an array of metabolic and homeostatic abnormalities, leading to neuronal dysfunction and death [88]. Moreover, a culture of NSC-derived neurons or differentiated SH-SY5Y neurons in a media where infected astrocytes were grown showed an increase in the apoptosis rate of these neurons. This further suggests that soluble factors released by astrocytes cause neuronal death [89].

In correlation with this, studies on infected human astrocytes and COVID-19 post-mortem brain samples revealed differential expression of proteins associated with glycolysis/gluconeogenesis, carbon metabolism, and the pentose phosphate pathway. Also, proteomic analysis observed decreased metabolites supporting neuronal metabolism and function [89]. Similarly, elevated gene expression related to DNA methylation, apoptosis, and neurodegeneration was observed, similar to those in conditions such as PD and amyotrophic lateral sclerosis (Table 1) (Figure 2) [88]. This further suggests that the neurological complications associated with SARS-CoV-2 could also be associated with astrocyte changes after infection.

Overall, microglia and astrocytes are greatly affected by SARS-CoV-2 infection; its effect on neuronal synapses could explain the high number of patients with neurological sequelae after the infection. This can be further supported by the damage inflicted on the BBB, allowing viral entry and recruitment of immune cells to the brain, triggering immune responses that possibly contribute to the overall pro-inflammatory state in the brain.

### 3.3. Influenza Virus

Influenza (IV) is the viral agent responsible for many respiratory tract infections worldwide [4]. Four influenza viruses, A, B, C, and D, have been identified. Currently, two influenza viruses are responsible for human disease: the influenza A (IAV) subtypes H1N1, H1N2, H2N2, H3N2, and influenza B (IBV) [4,92]. In this sense, the determination of these subtypes is given by differences in the structural proteins of the virus, hemagglutinin (HA) and neuraminidase (NA) [4]. IVs belong to the *Orthomyxoviridae* family, being the only members of their genus and an unassigned family [4]. These are enveloped, negative-sense, segmented-stranded RNA viruses [4]. Infections with influenza A or influenza B can generate mild respiratory symptoms, with clinical signs such as fever, sore throat, rhinitis, and cough [92]. However, in more severe cases, patients can also display pneumonia, which can be fatal, mainly for the elderly and infant populations [92]. Besides these manifestations in the respiratory tract, influenza viruses have also been associated with neurological complications such as seizures, encephalitis, and encephalopathies [93].

The immune response against influenza involves the secretion of pro-inflammatory cytokines such as TNF-α and IL-6 [8]. The secretion of these cytokines can lead to a poor exchange of gases due to damage to the alveolar-capillary membrane, leading to hypoxemia [8]. H1N1 infection stabilizes HIF-1α levels, and it has been shown that this transcription factor is involved in the secretion of pro-inflammatory cytokines and viral replication in lung epithelial cells [8]. Moreover, in vitro studies have shown that H1N1 infection also causes a nuclear translocation of HIF-1α in normoxia conditions [94]. On the other hand, a knockdown of HIF-1α in a human epithelial cell line has been shown to promote influenza replication [95]. This indicates that the role of HIF-1α has not been completely established in influenza infections. Hypoxia may also be associated with damage to the CNS due to the virus. In this line, studies have found a possible link between hypoxia, severe brain injury, and encephalopathy in patients infected with influenza [96,97].

Influenza receptors sialic acid linked to galactose through α-2,3 linkage (SA-α 2,3-Ga)l and SA-α 2,6-Gal have been described as homogenously distributed in the microglial cell line BV2 [98]. It has also been shown that influenza A viruses (IAV) require EGFR for efficient cell entry [99]. As stated above, this receptor is also expressed in microglia, which could make it an essential part of influenza A infection (Table 1) [54].

Moreover, studies conducted with influenza A in mice have shown that the infection promotes microglial activation, showing an increase in MHC I and II, CD80, and F4/80, accompanied by an increase in the mRNA levels of IFN-γ and IFN-β in the brain [100]. This correlates with changes in the gene expression of tight junction proteins such as claudin-5 and ZO-1 since type I and II interferons have been associated with immune cell entry through the BBB, suggesting an impairment in the function of this barrier (Table 1) (Figure 3) [100]. However, no significant differences were seen in the expression of chemokines, such as CXCL9 and CXCL10, responsible for leukocyte attraction in the CNS [100]. On the other hand, CD45^hi^ CD11b^+^ and CD45^hi^ CD11b^−^ cells were found in the brains of infected mice, suggesting an impairment in the BBB, allowing the entry of peripheral immune cells [100].

As mentioned above, microglia play a role in the synaptic processes of neurons, specifically through pruning excessive synapses [100]. After an influenza A infection, this process is dysregulated along with an alteration of neuronal morphology [100]. This was also associated with changes in the gene expression of brain-derived neurotrophic factor (*bdnf*) and nerve growth factor (*ngf*) at 21 dpi, where BDNF expression decreased and NGF increased (Table 1) [100]. BDNF and NGF are neurotrophins contributing to synaptic plasticity, brain development, and maintenance [9]. Moreover, viral infections have been shown to interfere with neurotrophin signaling pathways, leading to neuronal impairment and death [9]. It would be interesting to evaluate neurotrophin signaling impairment further to elucidate the role of IVs in CNS infections.

Studies conducted on the BV2 microglial cell line showed that the influenza pdm09 virus is capable of infecting and replicating in the BV2 cell line, promoting an increase in the secretion of IL-1β, IL-6, CCL2, and TNF-α [98]. However, this increase decreased 48 h after infection [98]. This cytokine secretion profile correlated with OPN levels in the infected BV cells (Table 1) (Figure 3) [98]. OPN, also known as secreted phosphoprotein 1, is an extracellular matrix glycoprotein that can be induced by influenza A infection and promote signaling through the PI3K, MAPK, and NF-κB pathways [98]. These pathways produce pro-inflammatory cytokines, virus proliferation, and OPN expression [98].

Studies conducted on isolated astrocytes from mice infected with H1N1 show an increase in the apoptotic rate of these cells in association with an activation of the apoptotic pathway dependent on caspase due to the viral infection [101]. This was determined by increases in the expression of the proteins caspase-3 and Bax (Table 1) (Figure 3) [101]. This suggests that targeting CNS damage caused by H1N1 can be done by regulating astrocyte apoptosis during infection [101]. The infection of the human differentiated astrocyte cell line T98G with H5N1 and H1N1 showed active virus transcription within the cells and the expression of influenza virus receptors SA-α 2,3-Gal and SA-α 2,6-Gal [102]. However, astrocytes also express EGFR, which could play a role in influenza A infection [53,99]. The H5N1 infection also promoted a high upregulation of IL-6 and TNF-α compared to the H1N1 infection, where only a slight increase was seen (Table 1) (Figure 1) [102]. These cytokines have been associated with acute viral-related encephalitis and encephalopathy cases [103].

On the other hand, an infection with the avian H7N9 virus also promoted a cytopathic effect after 24 h in a T98G cell line [103]. However, this was only minor in the case of the pandemic (pdm) H1N1 infection [103]. This difference also correlates with the detection of the viral M gene, where T98G cells infected with H7N9 show higher mRNA levels than those infected with pdmH1N1 [103]. Interestingly, the viral titer of H7N9 and pdmH1N1 was undetectable, which suggests that astrocytes do not support the viral production of these strains [103]. However, a detectable viral titer was seen 48 h after H5N1 infection [103]. Despite the lack of viral titer in the human astrocytes, the infection with H7N9 promoted an increased expression of pro-inflammatory cytokines, such as TNF-α, IL-6, IL-8, CCL2, and IFN-β 24 hpi (Table 1) (Figure 3) [103].

Since microglia and astrocytes express influenza receptors, further studies should evaluate if neurological complications associated with the infection arise independently of their expression. This could lead to new future therapies that control the pro-inflammatory state induced by microglia and astrocytes in response to influenza.

### 3.4. Human Parvoviruses

Within the human parvoviruses, we can find two pathogenic parvoviruses: The human parvovirus B19 (B19V) and the human bocavirus 1 (HBoV1) [7]. Parvoviruses are a family of small, non-enveloped viruses with a single-stranded DNA molecule [7]. They belong to the *Parvoviridae* family, the *Parvovirinae* subfamily, and the *Erythroparvovirus* and *Bocaparvovirus* genera, respectively [7].

B19V transmission is commonly through the respiratory route. Moreover, it has been described that this virus can be transmitted vertically and through the transfusion of blood products and bone marrow transplants [104]. Significantly, the clinical manifestations of B19V depend on host variables, including age, hematologic status, and immunological status [104]. This virus can rarely cause respiratory infections, but in children, it commonly causes erythema infectiosum and arthropathy in adults [104,105]. Moreover, B19V can cause hydrops fetalis and reach the bone marrow, causing pure red cell aplasia (PRCA) in immunocompromised hosts [7]. However, a higher prevalence of B19V-specific antibodies can be seen in adults over 18 [7]. It causes fever, malaise, myalgia, headaches, and other clinical manifestations such as cutaneous eruptions and arthralgia [7]. However, B19V infections are mostly asymptomatic, where seroconversion arises without an apparent illness [7]. In a few cases, respiratory compromise has been reported [106,107]. 

On the other hand, HBoV1 is a respiratory pathogen that is capable of infecting children around 5 to 15 years old. It is mainly found in children with upper respiratory tract infections [7]. It has a seroprevalence of 80% in six-year-olds, making it the most common human bocavirus [7]. The clinical manifestations of HBoV1 include hypoxia, pneumonia, bronchiolitis, and asthma exacerbations [7,8]. Hypoxia has also been shown to cause B19 replication and expression upregulation [108]. Moreover, it has been reported that HBoV1 can lead to viremia [7]. Importantly, parvoviruses can remain in tissues throughout the body after infection, making their study critical [109]. 

B19V and HBoV1 have also been implicated in neurological manifestations that rarely occur [7,110]. Patients infected with B19V have been shown to present symptoms such as encephalitis, ataxia, meningitis, meningoencephalitis, stroke, Guillain–Barré syndrome, and neuropathy [7,110,111]. B19V can remain latent after primary exposure in various organs, including the CNS [112]. Moreover, postmortem analyses of a fetus with increased B19V-specific IgG have shown B19V DNA in the nucleus of multinucleated giant cells and the frontal lobe endothelial cells [113]. Supporting this, PCR analyses conducted on HIV-infected and non-infected patients during autopsies detected B19V DNA in brain tissue (Figure 4) [114,115].

HBoV1 has also been linked in some rare cases to neurological manifestations, such as seizures, encephalitis, and encephalopathy [116,117,118,119]. Patients infected with the virus have also uncommonly presented convulsions and deterioration of consciousness problems [120]. Some of these patients also showed status epilepticus, encephalopathy, and even long-term cognitive and hearing problems [120]. Moreover, HBoV-specific IgG and IgM have been found in the CSF of children with severe or suspected viral encephalitis [117,118]. There has also been a case of HBoV1 in association with an acute necrotizing encephalopathy of a 3-year-old [119]. HboVs have also been linked to a possible role in triggering meningitis and meningoencephalitis (Figure 4) [121]. Neurological consequences of HBoV-1 have been associated with re-infections or re-activation of the virus [122]. This could be due to the latency or permanence of the virus on different tissues after infection, which could, in turn, activate pathways that induce neurological damage, such as the activation of the NLRP3 inflammasome, leading to cell death [123]. HBoV1 has been shown to activate DNA damage response (DDR) pathways without DNA damage, as was seen after transfection studies with HEK293 cells [123]. These pathways may also be activated in the brain, contributing to neurological sequelae. 

Lastly, studies conducted on canine parvovirus type 2 have shown brain lesions that could be associated with a hypoxic state induced by myocardial lesions in the canine [124]. Considering this, it is possible that the rare neurological manifestations linked to B19 and HBoV1 could be associated with hypoxia, which is also one of the main symptoms of these infections [8].

The receptors used by human parvoviruses to infect cells are still being studied. In the case of HBoV infection, the receptor used is still unknown [125]. On the other hand, globoside has been described as a receptor for B19V, and it has also been suggested that it infects the monocytic cell line U937 through an antibody-enhancement pathway [126]. Lastly, AXL, another receptor described for SARS-CoV-2, has also been identified as a co-receptor for B19V [71,127]. Since microglia and astrocytes express AXL, it could be suggested as a receptor for B19V in these cells (Table 1) (Figure 4) [74,91].

Studies with these parvoviruses and microglia or astrocytes are scarce. Regarding B19V, immunohistochemistry analyses of the brain tissue of patients with unspecified encephalopathy show more B19V-positive astrocytes in the frontal lobe compared to the control group [112]. However, microglial cells in the white matter, characteristic of encephalopathy, were B19V-negative [112]. Studies in brain samples positive for canine bocavirus 2 (CBoV-2) with encephalitis showed glial cell aggregates and gliosis [128]. This suggests a similar situation could occur with HBoV1 or B19V (Table 1) (Figure 4). Moreover, HBoV-1 plays an important role in co-infections with other viruses [129]. This could suggest that similar pathways are used for microglia, astrocytes, and neurological damage due to a pro-inflammatory response.

Despite the few studies on B19 and HBoV1, considering previous information on other respiratory viruses, neurological complications could also be mediated by microglia and astrocytes. However, further studies need to be conducted to determine this relationship.

**Table 1 microorganisms-12-01713-t001:** Effects of respiratory viruses on microglia and astrocytes.

Respiratory Virus	Parameter	Microglia	Astrocytes	References
hRSV	Receptor	CX3CR1? Nucleolin? EGFR? ICAM-1? HSPGs?	CX3CR1? Nucleolin? EGFR? ICAM-1? HSPGs?	[48,49,50,51,52,53,54,55,56]
	Cytokine secretion	IL-1β	IL-4 IL-10 CCL2 IL-6 TNF-α	[46,58]
	Effect	Microglial activation M1 polarization ROS production Neuronal death	Astrocyte activation Increase BBB permeability	[46,57,58,59]
SARS-CoV-2	Receptor	ACE2? CD147? HSPGs? AXL?	ACE2? NRP-1 Eph? CD147? HSPGs? AXL?	[22,55,56,71,72,73,74,87,88,91]
	Cytokine secretion	IL-1β IL-6 TNF-α	IL-6 TNF-α MCP1 MMP9 IL-17 CXCL10 CXCL6 CXCL1 CXCL2	[63,78,86,88]
	Effect	Microgliosis M1 polarization ER stress responses Apoptosis Synapsis elimination IFN signaling upregulation TLR2 activation Neurodegeneration?	Astrogliosis BBB disruption Translocation to the piriform cortex Morphological changes Downregulation of CAMK2D, ERBB2, C1QL, SYPL1 Upregulation of IFN-1 Neuron cell death Decrease in metabolites of neural metabolism and function Increase in gene expression of apoptosis, DNA methylation, and neurodegeneration	[63,64,75,79,80,84]
IVs	Receptor	SA-a 2,3-Gal SA-a 2,6-Gal EGFR?	SA-a 2,3-Gal SA-a 2,6-Gal EGFR?	[98,99,102]
	Cytokine secretion	IFN-γ IFN-β IL-1β IL-6 MCP-1 TNF-α	IL-6 TNF-α IL-8 CCL2 IFN-β	[98,100,102,103]
	Effect	Microglial activation Increase in MHC I/II, F4/80, CD80 BBB impairment? Dysregulation of pruning of excessive synapses Alteration of neuronal morphology Decrease in BDNF expression Increase in NGF and OPN expression	Increase in apoptotic rate Increase in expression of caspase 3 and Bax Cytopathic effect	[98,100,101]
Human Parvoviruses	Receptor	AXL?	AXL?	[74,91,125,126,127]
	Cytokine secretion	Unknown	Unknown	
	Effect	Gliosis? Glial cell aggregates?	Increase in B19V-positive astrocytes Gliosis? Glial cell aggregates?	[112,128]

## 4. Conclusions

Respiratory viruses have proven to threaten public health significantly [4]. They cause many symptoms, mainly within the respiratory tract, but also affect the CNS [4,7,9,10]. In this sense, viruses can provoke complications such as encephalitis, encephalopathies, and seizures, which have long-term consequences, such as schizophrenia, autism spectrum disorder, and mood disorders [4,7,9,10,11]. Within the CNS, microglia and astrocytes participate in various processes, one of them being mounting immune responses [3]. Importantly, viruses such as hRSV, SARS-CoV-2, IVs, B19V, and HBoV1 have been shown to promote clinical manifestations in the brain system [4,7,9,10]. However, the exact mechanisms by which this is accomplished are still unknown. Therefore, the implications of these respiratory viruses on microglia and astrocytes are a critical factor to consider in this study. After hRSV, SARS-CoV-2, and IVs infection, these glial cells activate, promoting the secretion of pro-inflammatory cytokines, activating apoptotic pathways, and interfering with correct neuronal function [22,46,57,58,59,63,64,75,76,79,80,82,84,86,88,89,98,100,101,102,103]. Regarding parvoviruses, their implications on microglia and astrocytes have not been thoroughly inspected. Further studies on glial cells after respiratory viral infections would help elucidate the effects of common respiratory viruses on the CNS. New knowledge is needed to prevent and treat neurological sequelae.

## Figures and Tables

**Figure 1 microorganisms-12-01713-f001:**
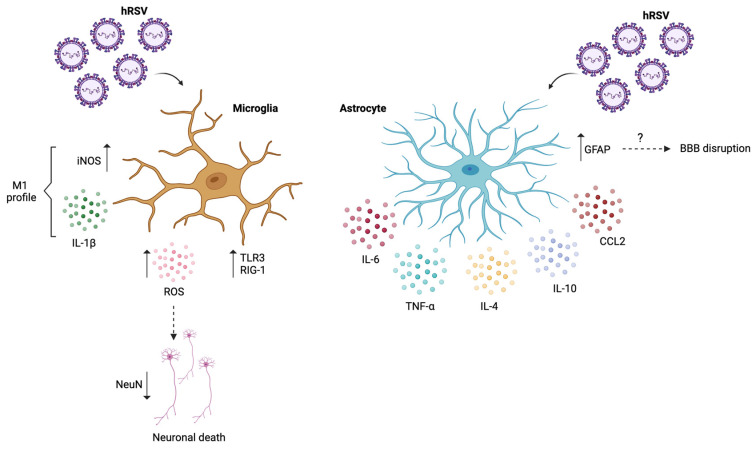
Effects of hRSV on microglia and astrocytes. The cell responses of microglia after hRSV infection indicate a polarization towards the M1 pro-inflammatory profile, alongside increases in TLR3 and RIG-1 expression. There is also an increase in ROS production, leading to neuronal death. The cell responses of astrocytes after hRSV infection led to an activation of astrocytes and BBB disruption by an unknown mechanism, accompanied by increases in the gene expression of IL-4, IL-10, and CCL2 and IL-6 and TNF-α production. Created with Biorender; Agreement #EL26KC08YQ.

**Figure 2 microorganisms-12-01713-f002:**
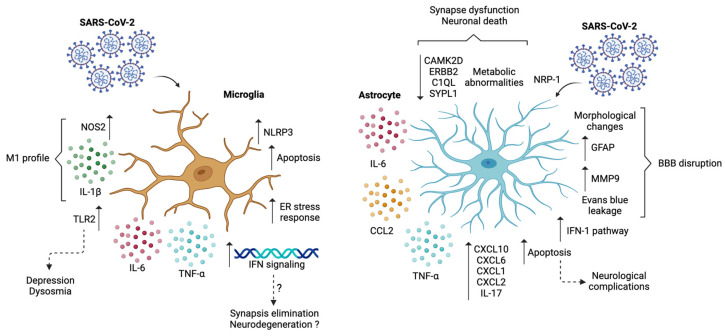
Effect of SARS-CoV-2 on microglia and astrocytes. The cell responses of microglia after SARS-CoV-2 infection indicate a polarization towards an M1 pro-inflammatory profile and an increase in apoptosis, ER stress response, NLRP3 inflammasome components, IFN signaling, possibly leading to synapsis elimination and neurodegeneration. The infection also causes TLR2 activation, which is associated with depression and dysosmia. The cell responses of astrocytes after SARS-CoV-2 infection led to an activation of astrocytes, an increase in MMP9, morphological changes, and evidence of Evans blue dye leakage, suggesting a BBB disruption. There is also an increase in the IFN-I pathway, CXCL10, CXCL6, CXCL2, CXCL1, and IL-17, alongside the secretion of pro-inflammatory components TNF-α, IL-6, and CCL2. There is an increase in apoptosis and metabolic abnormalities, alongside a decrease in CAMK2D, ERBB2, C1QL, and SYPL1, which suggests neurological complications, synapse dysfunction, and neuronal death. (Created with Biorender; Agreement number #TC26KC0CKS).

**Figure 3 microorganisms-12-01713-f003:**
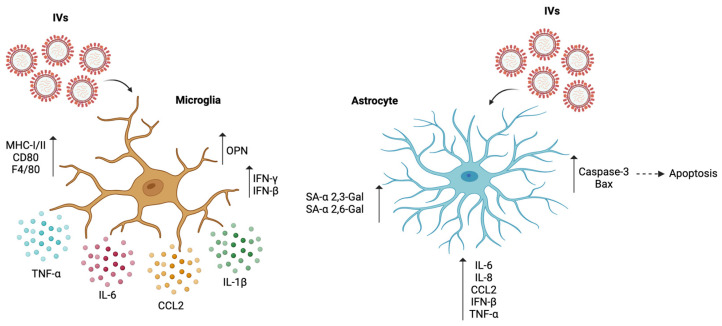
Effect of IVs on microglia and astrocytes. The cell responses of microglia after IVs infection indicate an increase in MHC-I, MHC-II, and F4/80, accompanied by a pro-inflammatory profile and the secretion of TNF-α, IL-6, CCL2, and IL-1β. There is also an increase in IFN-γ, IFN-β and OPN levels. The cell responses of astrocytes after IVs infection are characterized by an increase in the expression of caspase-3 and Bax, leading to an increase in apoptosis. It also shows a pro-inflammatory profile with increased IL-6, IL-8, CCL2, IFN-β and TNF-α. There is also an increase in the expression of IVs receptors SA-α 2,3-Gal and SA-α 2,6-Gal. (Created with Biorender; Agreement number #HF26KDBRMQ).

**Figure 4 microorganisms-12-01713-f004:**
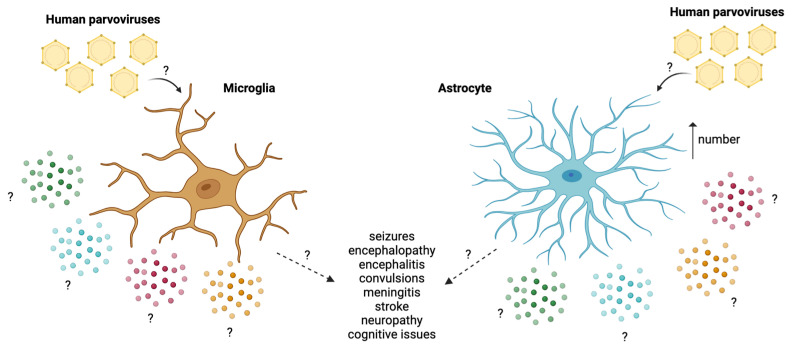
Effect of B19V and HBoV1 on microglia and astrocytes. It is unknown how parvoviruses interact with microglia and astrocytes, and different cytokines may be produced (interrogation points). It has been described that B19V-positive astrocytes increase after infection. Neurological sequelae associated with human parvoviruses include seizures, encephalopathy, encephalitis, convulsions, meningitis, stroke, neuropathy, and cognitive issues; however, it is unknown how B19V and HBoV1 cause these effects. (Created with Biorender; Agreement number #UN26KDEMUO).

## Data Availability

Not applicable.

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
