# Peer review of "Neurological Impact of Respiratory Viruses: Insights into Glial Cell Responses in the Central Nervous System"

_microorganisms, 2024, doi:10.3390/microorganisms12081713_

Round 1

Reviewer 1 Report

Comments and Suggestions for Authors

1. The manuscript offers a valuable compilation of existing literatures on the subject. However, it seems to primarily focus on collating information rather than engaging in the extensive analysis, critical evaluation, and synthesis commonly associated with review articles.

2. For section 3.1. Human Respiratory Syncytial Virus, it is challenging to grasp the main points the authors intended to convey.

Comments on the Quality of English Language

In some sections, the manuscript presents readability and clarity challenges. For instance, the phrase “Based on the importance of microglia and astrocytes in mounting immune responses in the brain. appears to be an incomplete sentence.

Author Response

Answer to Reviewer 1

1.- Reviewer 1: The manuscript offers a valuable compilation of existing literatures on the subject. However, it seems to primarily focus on collating information rather than engaging in the extensive analysis, critical evaluation, and synthesis commonly associated with review articles.

Answer: We would like to thank the reviewer for this comment. Further analysis and critical evaluation were added to the discussion. (Page 5; Lines 200-202, Page 6; Lines 229-231, 244-245 and 246-250, Page 10; Lines 437-441, Page 12; Lines 535-538, Page 14; Lines 603-606).

2.- Reviewer 1: For section 3.1. Human Respiratory Syncytial Virus, it is challenging to grasp the main points the authors intended to convey.

Answer: We would like to thank the reviewer for this comment. As requested, a final paragraph was added to clarify the main points of section 3.1. (Pages 6; Lines 244-248).

3.- Reviewer 1: Comments on the Quality of English Language

In some sections, the manuscript presents readability and clarity challenges. For instance, the phrase “Based on the importance of microglia and astrocytes in mounting immune responses in the brain.” appears to be an incomplete sentence.

Answer: We would like to thank the reviewer for this comment. As requested, the phrase “Based on the importance of microglia and astrocytes in mounting immune responses in the brain.” was modified and re-written to clarify the sentence. (Page 4; Lines 155-159). Also, the manuscript was carefully revised to improve readability and clarity. (Page 1; Lines 16, 24 and 32, Page 2; Lines 51, 60, 65, 69, 73, 80, and 91, Page 3; Lines 102, 103, 108, 110, 117, 131, 135, 136, Page 4; Lines 143, 149, 161, 169, 185, Page 5; Lines 203, 207, Page 6; Lines 234, 237, Page 7; Lines 279, 288-290, 292, 293, 294, 315 and 316, Page 8; Lines 330, 340, 341, 344, 347 and 351, Page 10; Lines 408, 409, 417, 420, 445, 450, Page 11; Lines 470, 473, 474 and 481 Page 12; Lines 495, 499, 509, 516, 519, 520, 525-527, Page 14; Lines 577, 601 and table (references), Page 16; Line 611).

Reviewer 2 Report

Comments and Suggestions for Authors

It would be interesting if the authors wrote a review comparing the effect of respiratory viruses on the nervous system with that of strictly neurotropic viruses. Excellent paper.

Author Response

Answer to Reviewer 2

1.- Reviewer 2: It would be interesting if the authors wrote a review comparing the effect of respiratory viruses on the nervous system with that of strictly neurotropic viruses. Excellent paper.

Answer: We would like to thank the reviewer for this comment. We appreciate your suggestion for a further review article comparing the effects of respiratory and neurotropic viruses.

Reviewer 3 Report

Comments and Suggestions for Authors

The authors have demonstrated a commendable grasp of the intricate mechanisms underlying the impact of various viruses on the Central Nervous System (CNS), with a particular emphasis on respiratory viruses. The exhaustive review of the host-viral interaction and the detailed description of the pathogenesis involved is indeed noteworthy. However, a critical point concerning respiratory infection appears to have been overlooked. The authors' exclusive reliance on the pathogen's and host's molecular virology does not solely determine the clinical implications. Respiratory infection mediates reduced oxygen supply to the systemic organs, including the brain, liver, kidney, and spleen. It is, therefore, essential to consider the CNS signs that may result from hypoxia and hypoxia-related disease.

Furthermore, it is essential to consider the overall perspective of the host rather than solely focusing on the pathogen. The host is a complex organism that communicates through various means, such as hormones, chemokines, and eicosanoids, thereby influencing all other organs. This aspect should be incorporated into the analysis to provide an all-encompassing understanding of the impact of respiratory viruses on the CNS.

Comments on the Quality of English Language

The English quality is fine, with only minor editing required.

Author Response

Answer to Reviewer 3

1.- Reviewer 3: The authors have demonstrated a commendable grasp of the intricate mechanisms underlying the impact of various viruses on the Central Nervous System (CNS), with a particular emphasis on respiratory viruses. The exhaustive review of the host-viral interaction and the detailed description of the pathogenesis involved is indeed noteworthy. However, a critical point concerning respiratory infection appears to have been overlooked. The authors' exclusive reliance on the pathogen's and host's molecular virology does not solely determine the clinical implications. Respiratory infection mediates reduced oxygen supply to the systemic organs, including the brain, liver, kidney, and spleen. It is, therefore, essential to consider the CNS signs that may result from hypoxia and hypoxia-related disease.

Answer: We would like to thank the reviewer for this comment. Hypoxia effects were considered on the respiratory and CNS manifestations of the viral pathogens described. (Page 1; Lines 41-44, Page 4; Lines 169-175 and 178-184, Page 5; Lines 209-211, Page 6-7; Lines 264-277, 282-286, and 307-309, Page 9; Lines 365-367, Page 11; Lines 456-468, Page 13; Lines 554-556, Page 14; Lines 583-587).

2.- Reviewer 3: Furthermore, it is essential to consider the overall perspective of the host rather than solely focusing on the pathogen. The host is a complex organism that communicates through various means, such as hormones, chemokines, and eicosanoids, thereby influencing all other organs. This aspect should be incorporated into the analysis to provide an all-encompassing understanding of the impact of respiratory viruses on the CNS.

Answer: We would like to thank the reviewer for this comment. A deeper analysis of the impact of respiratory viruses and CNS was incorporated. (Page 5; Lines 200-202, Page 6; Lines 229-231, 244-245 and 246-250, Page 10; Lines 437-441, Page 12; Lines 535-538, Page 14; Lines 603-606).

3.- Reviewer 3: Comments on the Quality of English Language

The English quality is fine, with only minor editing required.

Answer: We would like to thank the reviewer for this comment. To evaluate the quality of the English language the manuscript was revised, and editing was carried out where needed (Page 1; Lines 16, 24 and 32, Page 2; Lines 51, 60, 65, 69, 73, 80, and 91, Page 3; Lines 102, 103, 108, 110, 117, 131, 135, 136, Page 4; Lines 143, 149, 161, 169, 185, Page 5; Lines 203, 207, Page 6; Lines 234, 237, Page 7; Lines 279, 288-290, 292, 293, 294, 315 and 316, Page 8; Lines 330, 340, 341, 344, 347 and 351, Page 10; Lines 408, 409, 417, 420, 445, 450, Page 11; Lines 470, 473, 474 and 481 Page 12; Lines 495, 499, 509, 516, 519, 520, 525-527, Page 14; Lines 577, 601 and table (references), Page 16; Line 611).

Round 2

Reviewer 1 Report

Comments and Suggestions for Authors

The paper focuses on microglia and astrocytes but neglects the role of oligodendrocytes, another critical type of glial cell in the CNS. It creates a gap in the comprehensive understanding of glial cell responses to respiratory viruses.

Some sections rely on older references, missing out on recent advancements in the field. For example, the discussion on SARS-CoV-2 could be enhanced with more recent studies that provide updated insights into the virus's mechanisms.

Author Response

Answer to Reviewer 1

1.- Reviewer 1: The paper focuses on microglia and astrocytes but neglects the role of oligodendrocytes, another critical type of glial cell in the CNS. It creates a gap in the comprehensive understanding of glial cell responses to respiratory viruses.

 Answer: We would like to thank the reviewer for this comment. We briefly mentioned the role of oligodendrocytes as a type of glial cell in the CNS. However, we focused the review on microglia and astrocytes due to their importance in mounting immune responses in the brain. (Page 2; Lines 61-64).

2.- Reviewer 1: Some sections rely on older references, missing out on recent advancements in the field. For example, the discussion on SARS-CoV-2 could be enhanced with more recent studies that provide updated insights into the virus's mechanisms.

Answer: We would like to thank the reviewer for this comment. We added more recent literature in our discussion shedding light on possible mechanisms (Page 7; Lines 281, 292, 304-306, 311-313, Page 8; Lines 346-349, Page 9; Lines 382-386, Page 16, Lines 635, 636).